# Neo-5,22*E*-Cholestadienol Derivatives from *Buthus martensii* Karsch and Targeted Bactericidal Action Mechanisms

**DOI:** 10.3390/molecules24010072

**Published:** 2018-12-26

**Authors:** Biyu Lv, Weiping Yin, Jiayu Gao, Huaqing Liu, Kun Liu, Jie Bai, Qiangqiang Yang

**Affiliations:** 1School of Chemical Engineering and Pharmaceutics, Henan University of Science and Technology, Luoyang 471023, China; biyulv0827@163.com (B.L.); cruise1024@163.com (J.G.); lhq@126.com (H.L.); liukun1988china@163.com (K.L.); bj@126.com (J.B.); 2Life Science and Environmental Science Research Center of Harbin University of Commerce, Harbin 150025, China; 18568105396@163.com

**Keywords:** *Buthus martensii* karsch, secondary metabolites, novel 5,22*E*-cholestadienol compounds, broad-spectrum bactericide, natural products, structures elucidated, molecular docking, targeted action mechanisms

## Abstract

The discovery and search for new antimicrobial molecules from insects and animals that live in polluted environments is a very important step in the scientific search for solutions to the current problem of antibiotic resistance. Previously, we have reported that the secondary metabolite with the antibacterial action discovered in scorpion. The current study further isolated three new compounds from *Buthus martensii* karsch, while compounds **1** and **2** possessed 5,22*E*-cholestadienol derivatives whose structure demonstrated broad spectrum bactericide activities. To explore the antibacterial properties of these new compounds, the result shows that compound **2** inhibited bacterial growth of both *S. aureus* and *P. aeruginosa* in a bactericidal rather than a bacteriostatic manner (MBC/MIC ratio ≤ 2). Similarly, with compound **1**, a ratio of MBC/MIC ≤ 2 indicates bactericidal activity inhibited bacterial growth of *P. aeruginosa.* Remarkably, this suggests that two compounds can be classified as bactericidal agents against broad spectrum bactericide activities for 5,22*E*-cholestadienol derivatives from *Buthus martensii* karsch. The structures of compounds **1**–**3** were established by comprehensive spectra analysis including two-dimensional nuclear magnetic resonance (2D-NMR) and high-resolution electrospray ionization-mass spectrometry (HRESI-MS) spectra. The antibacterial mechanism is the specific binding (various of bonding forces between molecules) using compound **1** or **2** as a ligand based on the different receptor proteins’—2XRL or 1Q23—active sites from bacterial ribosome unit A, and thus prevent the synthesis of bacterial proteins. This unique mechanism avoids the cross-resistance issues of other antibacterial drugs.

## 1. Introduction 

The scorpion is an ancient arthropod and a source of bioactive functional ingredients, it has been traditionally long time used for medicinal benefits as analgesic and antitumor effects of scorpion toxin in traditional Chinese medicine (TCM). The secondary metabolites of insects and arthropods are an important resource for the study of natural products [1,2]. Insects, such as mosquito, flies, and bedbugs, adapted under the harshest living conditions on earth and developed bioactive secondary metabolites responding to environmental challenges [3]. However, due to the low content levels and difficulty of separation, the assessment of the diversity of active molecules and the identification of novel structures small molecular remain a continuing challenge. Previously, the secondary metabolites with antibacterial features have been isolated from insects, invertebrates and arthropod in our group [4,5,6]. It especially highlights the antibiotic potentials of secondary metabolites, including low-abundance analogs in natural products. The recent study investigated the antibacterial effects of the scorpion, *Buthus martensii* karsch and its secondary metabolites, thus three compounds (Figure 1) were firstly isolated and identified. Among them, two compounds deduced as the novel 5,22*E*-cholestadienol derivatives were determined to have the bactericidal abilities.

## 2. Results and Discussion 

### 2.1. Chemistry

The bioassay guided isolation using silica gel and preparative HPLC gave compounds **1** (35.0 mg), **2** (29.1 mg), and **3** (24.6 mg). 

Compound **1** (QX75-5) was obtained as a white solid powder and assigned a molecular formula for C_27_H_43_O_2_ based on its positive HR-ESIMS ion at [M + H]^+^
*m*/*z* 399.3263 (calcd. for 399.3258). Optical rotation of 1 was measured [α${]}_{D}^{25}$ −22.7° (c, 0.07, CHCl_3_) by a JASCO DIP-360 digital polarimeter. The IR spectrum of compound 1 suggested the presence of α,β-unsaturated carbonyl group (1678 cm^−1^) and the double bond (1583, 1446, and 690 cm^−1^), a hydroxyl absorption (3346 and 1110 cm^−1^). The ^1^H-NMR (400 MHz, CDCl_3_) spectrum exhibited resonances multiple proton signals for the characteristics of sterols or triterpenoid compounds at δ*_H_* 0.68–2.52, besides which it can be seen the signal of a hydroxyl at δ*_H_* 3.62, and the double bonds at δ*_H_* 5.69 (1H, d, 1.48 Hz), δ*_H_* 5.68 (1H, dd, 3.24, 1.48 Hz), and 5.20 (1H, dd, 7.60, 3.28 Hz) suggesting that compound **1** is a steroidalenol structure with α,β-unsaturated carbonyl group combining the above IR data. The ^13^C-NMR and DEPT (100 MHz, CDCl_3_) spectra (Table 1) displayed resonances for 27 carbon signals and categorized as unsaturated double bond at δ*c* 135.6 (CH), 131.9 (CH), 126.1 (CH); five methyl at δ*_C_* 12.0, 17.3, 18.9, 22.7, and 22.8; four quaternary carbon signals at δ*c* 202.3 (C=O), 165.1 (C), 70.5 (C), and 42.3 (C); tertiary carbon of six signal δ*c* 54.8 (CH), 50.0 (CH), 45.4 (CH), 42.8 (CH), 36.3 (CH), and 29.7 (CH); alone with nine methylene carbon signals δ*c* 39.5 (CH_2_ × 2), 38.7 (CH_2_), 38.3 (CH_2_), 35.7 (CH_2_), 31.2 (CH_2_), 28.1 (CH_2_), 23.8 (CH_2_), and 21.2 (CH_2_). The observed HMBC correlations from H-4 (δ*_H_* 5.68) to C-3 (δ*_C_* 202.3) C-4 (δ*_C_* 126.1), and C-5 (δ*_C_* 165.1), H-22 (δ*_H_* 5.22), to C-23 (δ*_C_* 135.6), certified the presence of two double bonds at C-4 and C-22, respectively. In addition, the HMBC correlations from H-17 (δ*_H_* 1.58) to C-12 (δ_C_ 39.5) and C-21 (δ*_C_* 18.9), H-23 (δ*_H_* 5.20, 1H, dd, 7.60, 3.28 Hz) to C-25 (δ_C_ 70.5), revealed the β-configuration of C-21 methyl and one hydroxyl group joined on the C-25. The HMBC correlation of compound **1** is shown in Figure 2. From the above, compound **1** was confirmed as a novel cholesterolenone structure [7,8] based on the analysis of spectral data of **1** and the comparison with the literature. Compound **1** was then named as (−) 22*E*, 3-oxocholesta-4, 22 (23)-dien-25-ol according to reported compound in the literature (+) (22*E*, 24α)-24-Ethyl-3-oxocholesta-4, 22 (23)-dien-25-ol [9], which the latter for known compound has one ethyl group former on C-24 (for details in Figure 3).

Since the optical rotation [α${]}_{D}^{25}$ −22.7° of compound **1** exhibited a negative Cotton effect at 223.5 nm for the determination of its CD spectra, and it was inferred to be a levorotatory configuration compound of enantiomers contrasted with known compound (+)7*S*,8*S*,9*R*,13*R*,14*S*,17*R*,24*S*,20*R* (22*E*,24α)-24-Ethyl-3-oxocholesta-4, 22(23)-dien-25-ol [9]. Accordingly, the structure of compound **1** was elucidated as a new cholestadienol shown and named as (−) 7*S*, 8*S*, 9*R*, 13*R*, 14*S*, 17*R*, 20*S* (22*E*)-3-oxocholesta-4, 22(23)-dien-25-ol in Figure 1. Which the absolute stereochemistry can be seen from the NOESY spectrum, and the correlation crossover was at H-17ax (δ*_H_* 1.58), H-12ax (δ*_H_* 1.96), and H-20eq (δ*_H_* 1.18), from Me-21 to H-22 (δ*_H_* 5.22); both H-17ax and H-20eq (^3^*J*_H17–H20_) with 3.78, 1.36 Hz in ^1^H-NMR were indicative of the 17*R*, 20*S*, and 21*R* configuration. Their absolute configurations were confirmed by comparing with known chiral analogs, C-24*R* orientation, and the *J* values above analysis of chemical correlation in ^1^H-NMR spectrum. The ^1^H-NMR and ^13^C-NMR data assignment of compound **1** was shown in Table 1. 

The HR-EIMS of compound **2** (QX37-45-4) exhibited M + H^+^ at *m*/*z* 427.3568, which corresponded to the formula C_29_H_47_O_2_ (calcd. for 427.3571). It was a yellowish amorphous solid with optical rotation measured as [α${]}_{D}^{25}$ −66.7° (c, 0.10, CHCl_3_). The CD spectrum of compound **2** exhibited a negative Cotton effect at 222.5 and 254.8 nm (vw) (Figure 3), and its IR absorption bands at 3417 cm^−1^ and 1726 cm^−1^ and 1640, 1568, and 1447 cm^−1^ coincided with the double bond and ester group, respectively. The ^1^H-NMR (400 MHz, CDCl_3_) and ^13^C-NMR and DEPT (100 MHz, CDCl_3_) spectra (Table 1) were similar to those of compound **1**, implying that compound **2** was a derivative of the same skeleton of 1 with a ketone group replaced by ester group of 3β-acetoxyl group [H-3, (δ*_H_* 3.50, 1H, m); C-3, (δ*_C_* 71.8, CH) ]. There are the double bond signals in compound **1** with an isolated double bond at δ*_H_* 5.33 H-5 (1H, dd, 5.16 Hz, 3.24 Hz) and δ*_C_* 121.7 (C-5, CH), and δ*_C_* 140.7 (C-4, C); and two others double bond signals both δ*_H_* 5.35 H-22 (1H, dd, 3.56, 1.64 Hz)] and δ*_H_* 5.16 H-23 (1H, dd, 6.68, 3.56 Hz). Then, it was observed for corresponding carbon signals in its ^13^C-NMR (DEPT) spectrum at δ*_C_* 135.8 (C-22, CH) and δ*_C_* 131.7 (C-23, CH), respectively. The ^13^C-NMR spectra of compound exhibited 29 carbon signals including six methyls, nine methylenes, seven methenyl and three olefinic carbons, three quaternary carbon atoms, and one ester carbonyl carbon. Compound **2** was then identified as 3β-acetate,5(6)-22(23)-cholestadien by comparison of spectroscopic data in literature [10]. For the optical rotations and ^1^H-NMR spectrum of the metabolite 2 from *Buthus martensii* kirsch, that it was in agreement with the chemical structures of the corresponding CAS Registry Number 1089664-70-9 of SciFinder Scholar [11]. Therefore, the absolute configuration of compound **2** should be (−) 7*S*, 8*S*, 9*R*, 13*R*, 14*S*, 17*R*, 20*S*-(22*E*)-3β-acetate, 5(6)-22*E*-(23)-cholestadien, shown in Figure 4. The ^1^H-NMR and ^13^C-NMR data assignment of compound **2** was shown in Table 1. 

Compound **3** (QX3-2-7) was isolated as a colorless oil, [α${]}_{D}^{25}$ = −15.2° (c, 0.10, CHCl_3_). The molecular formula was determined as C_15_H_25_O_3_ from HR-ESIMS at *m*/*z* 253.1796 [M + H]^+^ (calcd. for 253.1798). IR spectrum suggested the presence of carboxyl acid for carbonoxyl group (1721 cm^−1^), double bond (1642 and 1460 cm^−1^) and hydroxyl group (3300–2500 and 1327 cm^−1^). The ^1^H-NMR (400 MHz, CDCl_3_) spectrum exhibited that the compound has two methyl signals both 0.87 (3H, s) and 0.88(3H, s), saturated protons signals in high field between 1.26 and 2.30. In addition, an unsaturated protons signal at δ*_H_* 5.34 (br, 1H) and one activity hydrogen signal was at δ*_H_* 7.54 (br, 1H). The ^1^H-^1^H COSY of compound 3 revealed the cross correlation signals of two spin systems between H-2/ H-3, H-2′, H-3′, H-4′ (δ*_H_* 5.34/2.01/1.26/2.30), and H-7/H-6, H-5 (δ*_H_* 0.87/1.62/2.01), indicating the existence of two substituents of the methyl group and saturated chains hydrocarbon on the ring (in Figure 5.) The ^13^C-NMR and DEPT (100 MHz, CDCl_3_) spectra (Table 2) displayed resonances for 15 carbon signals categorized as two methyls as δ*_C_* 14.0 (C-7) and 14.1 (C-1′), one double bond signal at δ*_C_* 130.0 (C-1) and 129.7 (C-2), a saturated carboxylic acid signal at δ*_C_* 179.1, and a quaternary-oxygen carbon signal at δ*_C_* 77.2 (C-6), another nine carbon signals at δ*_C_* 22.7 (CH_2_), 24.9 (CH_2_), 27.2 (CH_2_), 29.1 (CH), 29.3 (C), 29.5 (CH_2_), 29.7 (CH_2_), 32.0 (CH_2_), and 34.1 (CH_2_). The above mentioned data indicated that compound 3 possessed a α-bisaboleneol skeleton structure [12,13]. It can be inferred that a monocyclic terpene alcohol compound with a saturated carboxylic acid substituent and a hydroxyl group with active hydrogen was located at C-6, which correlation cross-signal δ*_H_* 5.34/δ*c* 77.2 was observed in the HSQC and HMBC spectrum, respectively; along with the ESI-HRMS indicated that structure of compound **3** was confirmed. The occurrence of a 6-hydroxy-7-methyl monocyclic terpene-1(2) moiety was demonstrated by the HMBC correlations from H-7 (δ*_H_* 0.87) to C-6 (δ*c* 77.2), C-2 (δ*c* 129.7), and C-3 (δ*c* 27.2), and from H-2 (δ*_H_* 5.34) to C-1 (δ*c* 130.0), C-3 (δ*c* 27.2). The saturated carboxylic group of substituent located at C-4 of the β-bisaboleneol ring, of which was substantiated by the observed HMBC cross-peaks of H-2′ (δ*_H_* 1.26) to C-4 (δ*c* 29.5), from H-3′ (δ*_H_* 2.30) to C-5(δ*c* 24.9), C-4′ (δ*c* 32.0) and from H-7′ (δ*_H_* 1.26) to C-5′ (δ*c* 29.5) and C-4′ (δ*c* 32.0). Therefore, the ^1^H-NMR and ^13^C-NMR data assignment of compound **3** was shown in Table 2. The ^1^H-^1^H COSY correlations and key HMBC correlations for compound **3** shown in Figure 5 and Figure 6, respectively.

Therefore, the planar structure of **3** was elucidated as a α-bisaboleneol acid and was reasonably assigned as 1β-methyl-6-hydroxyl-bicyclo[4,2,0]-hex-1-ene-4α-(2′-methyl)-isooctanoic acid. Analysis of the previously reported bisaboleneol acid revealed that almost all possessed hydroxyl group on ring, and the carboxyl group instead of the end of the carbon chain. However, it shows that compound **3** is a new compound isolated from the arthropods *Buthus martensii* kirsch, and has not been reported in the literature before.

The absolute configuration of **3** was established by comparison of the NOESY experimental with reference to the literature [14] and its CD spectra determined. The NOESY correlations of H-5eq/H-2′eq /H-4′, and H-5a/H-3eq (Figure 7a), as well as the coupling constants of H-5 and H-2′ value <10.0 Hz respectively in ^1^H-NMR spectrum in CDCl_3_ (Table 3) suggested *trans*-relationship for all of the ortho-position chiral centers. The absolute configurations (Figure 7b) of compound **3** were determined as (4*S*, 6*R*) by the close agreement between the references and experimental CD spectrum for 223.5 nm and 250.2 nm of compound **3**. Thus, compound **3** was proved to be a left-handed molecule and named as (−)-1β-methyl-6(*R*)-hydroxyl-bicyclo-[4,2,0]-hex-1-ene-4α(*S*)-(2′-methyl)-isooctanoic (Figure 7).

### 2.2. Assay of Antibacterial and Bactericidal Activities

The isolated compounds were tested for their antibacterial and germicide activities against four opportunistic pathogen strains—*Bacillus subtilis* ATCC 6051, *Staphylococcus aureus* ATCC 6538, *Escherichia coli* ATCC 25922 and *Pseudomonas aeruginosa* ATCC 27853—as well as methicillin-resistant *Staphylococcus aureus* (MRSA) by the two fold serial dilution method [15]. The positive control, penicillin G (potassium salt, 1598 units per mg) gentamycin (sulfate 1000 units per mg), and vancomycin (50 units per mg) were purchased from Sigma-Aldrich Ltd., St. Louis, MO, USA.

The results indicated that compounds **1** and **2**, both with 5,22*E*-cholestadienol structure, have a broad antibacterial spectrum against the opportunistic pathogen *P. aeruginosa* ATCC 27853 and *S. aureus* ATCC 6538. The minimal inhibitory concentrations (MICs) value of the sample was firstly measured by the dual-dilution method. In order to obtain the effective fungicide leading compound and further determine the MBC value, we selected the MIC value of less than 64 μg/mL for sustainable development and the testing sample. The results showed the significant antibacterial effect for the MIC values of 64 μg/mL of compound **1** against *P. aeruginosa* ATCC 27853 and 78 μg/mL against *S. aureus* ATCC 6538. Compound **2** displayed the stronger antibacterial effect with the MIC values of 16 μg/mL against both *P. aeruginosa* ATCC 27853 and *S. aureus* ATCC 6538. Then, the MBC values of 2 were at 32 μg/mL and less than 48 μg/mL, respectively. 5,22*E*-cholestadienol derivatives (compounds **1** and **2**) are small molecules of natural products with the significant inhibitory effects against pathogenic bacteria *S. aureus* ATCC 6538 and *P. aeruginosa* ATCC 27853. And MBC/MIC ratios of two compounds were quantified by using a luciferase-based assay [16]. According to the ratio of MBC/MIC, it is possible to identify the antibacterial profile of compounds (bacteriostatic and/or bactericidal). The result shows that compound **2** inhibited bacterial growth of both *S. aureus* and *P. aeruginosa* in a bactericidal rather than a bacteriostatic manner (MBC/MIC ratio ≤ 2). Similarly, with compound **1**, a ratio of MBC/MIC ≤ 2 indicates bactericidal activity inhibited bacterial growth of *P. aeruginosa*, whereas a ratio of MBC/MIC ≥ 4 defined a bacteriostatic effect [17]. Interestingly, this suggests that these two compounds with 5,22*E*-cholestadienol structure classified as bactericidal agents against broad spectrum bactericide activities from *Buthus martensii* karsch, which is a more interesting profile (Table 3). It was thus suggested that compound **2** had the most potent broad-spectrum bactericidal activity and need further investigation into the mechanistic insight. All in vitro experiments were performed in duplicate.

Compound **1** (QX75-5): (−)7*S*, 8*S*, 9*R*, 13*R*, 14*S*, 17*R*, 20*S* (22*E*)-3-oxocholesta-4,22(23)-dien-25-ol. [α${]}_{D}^{20}$ −22.7° (*c*, 0.07, CHCl_3_); UV (CHCl_3_) λ_max_: 223.5 nm; CD (*c,* 6.60 × 10^−4^ M, CHCl_3_) λ_max_ (Δε): 201 (−9.23), 205 (−13.5), 208 (−15.3), 210 (−14.1), 214 (−12.3), 219 (−10.5), 223 (−9.420), 229 (−8.56), 233 (−7.86). Positive HR-ESIMS *m*/*z* [M + H]^+^ 399.3263 (calcd. for 399.3258, C_27_H_43_O_2_). IR ^film^
_cm−1_: 3346, 1678, 1583, 1446, 690. For ^1^H-NMR and ^13^C-NMR data, see Table 1.

Compound **2** (QX37-45-4): (−) 7*S*, 8*S*, 9*R*, 13*R*, 14*S*, 17*R*, 20*S* (22*E*)-3β-acetate,5 (6)-22*E* (23)-Cholestadien. [α${]}_{D}^{20}$ −66.7° (*c*, 0.10, CHCl_3_); UV (CHCl_3_) λ_max_: 222.5 and 254.8 nm (vw); CD (*c,* 6.67 × 10^−4^ M, CHCl_3_) λ_max_(Δε): 204 (−5.84), 209 (−12.7), 213 (−10.1), 222.5 (−8.50), 230 (−6.54), 236 (−4.79) and 242 (−2.75), 247 (−1.44), 254 (−0.94), 260 (−1.16), 268 (−1.94). Positive HR-ESIMS *m*/*z* [M + H]^+^ 443.3153 (calcd. for 443.3156, C_29_H_47_O_2_). IR ^film^
_cm−1_: 3417, 1726, 1640, 1568, 1447. For ^1^H-NMR and ^13^C-NMR data, see Table 1. 

Compound **3** (QX3-2-7): (−) 1β-methyl-6 (*R*) -hydroxyl-bicyclo [4,2,0] hex-1-ene-4α (*S*)-(2′-methyl)-isooctanoic. [α${]}_{D}^{20}$ −15.2° (*c*, 0.10, CHCl_3_); UV (CHCl_3_) λ_max_: 223.5 and 250.2 nm (vw); CD (*c*, 6.61 × 10^−4^ M, CHCl_3_) λ_max_ (Δε): 201 (+4.15), 206 (−9.24), 219 (−10.0), 223 (−10.5), 227 (−9.60), 231 (−7.92), 236 (−6.10) and 244 (−4.16), 250 (−3.06), 260 (−2.30). Positive HR-ESIMS *m*/*z* [M + H]^+^ 253.1796 [M + H]^+^ (calcd. for 253.1798, C_15_H_25_O_3_). IR ^film^
_cm−1_: 3300–2500, 1721, 1642, 1460, 1327. The ^1^H-NMR and ^13^C-NMR data, see Table 2.

Appendix A can include anything data of reported compounds and they have been designated open access or are freely available online.

## 3. Materials and Methods 

### 3.1. General Experimental Procedure

Optical rotations were measured using a JASCO DIP-360 (Tokyo, Japan automatic digital polarimeter). IR and UV spectra were recorded using the JASCOFT-IR 620 spectrophotometer and UV-2600 instrument, respectively. 1D and 2D NMR spectra were recorded on Bruker DRX-400 spectrometer (400 MHz for ^1^H-NMR, Karlsruhe, Germany) with TMS used as internal standard. The mass spectra were obtained on Agilent Series1100 SL mass spectrometer (Agilent Technologies Inc., Santa Clara, CA, USA), and Bruker Daltonics mass spectrometer (Bruker Daltonics Inc. Billerica, MA, USA) with an electrospray ionization source. Circular Dichroism (CD) was obtained using the Chirascan, Applied Photophysics Ltd. (Surrey, UK). HPLC was performed using a system comprised of a CCPM pump (Tosoh, Tokyo, Japan), a CCP PX-8010 controller (Tosoh), an RI-8010 detector (Tosoh) or a Shodex OR-2 detector (Showa-Denko, Tokyo, Japan), and a Rheodyne injection port. A Capcell Pak C18 UG120 column (10 mm i.d. × 250 mm, 5 μm, Shiseido, Tokyo, Japan) was employed for preparative HPLC. Sephadex LH-20 (GE Healthcare Bio-Sciences AB, Uppsala, SE, USA) was used for column chromatography (CC), and a silica gel GF_254_ (10–40 mm, Haiyang Co., Qingdao, China) was used for preparative TLC as precoated plates. TLC spots were visualized under UV light through dipping into 5% H_2_SO_4_ in alcohol. All chemicals used were analytical grade.

### 3.2. Insect/Animal Materials

Adult of arthropod scorpions, *Buthus martensii* karsch (2.0 kg), were purchased from traditional a Chinese medicine store in Bozhou, Anhui of China in September 2016 and the insect has been identified by Professor Wenliang Li. Whole scorpion *B. martensii* karsch specimen no. QX2016-9 is currently stored in the FuniuMountain Natural Products Resource exhibition hall of Henan University Science and Technology, Luoyang city. 

In the antibacterial experiment, four selected pathogenic strains and one resistant strain were commonly used in clinic, including two Gram-positive bacteria (*B. subtilis* and *S. aureus*), two Gram-negative bacteria (*E. coli* and *P. aeruginosa*), and the multidrug-resistant strain (methicillin-resistant *Staphylococcus aureus*—MRSA). Four species of standard strains were acquired from Shanghai Biological Research Technology Co., Ltd. (Shanghai, China) and one species of MRSA strains was obtained from clinically isolated drug-resistant strain.

### 3.3. Extraction and Isolation 

Air-dried bodies of the scorpion *B. martensii* karsch were crushed into granules and extracted with CH_2_Cl_2_ to remove insect wax. Using the ultrasonic extraction method, the 70% methanol extract (76.00 g) was collected and freeze-dried. The extract was further dissolved in methanol and then separated solid from liquid to obtain methanol extract (52.10 g). Under the guidance of antimicrobial activity, the MeOH extracts were then concentrated and isolated by the silica gel column and Sephadex LH-20 column chromatograph with gradient eluted using CHCl_3_:MeOH (10:0, 9:1, 7:3, 5:5, 3:7; 1:9, 0:10) and MeOH:H_2_O. The same fractions were collected to produce Sections I–VI. Section II was purified repeatedly (CHCl_3_:MeOH, 9:1) by silica gel column as well as Sephadex LH-20 to produce compound **3** (24.62 mg). Section IV was further separated (CHCl_3_:MeOH, 7:3, 1.44 g) by preparative HPLC Column: YMC-ODS-AQ (250 cm × 20 cm); mobile phase MeOH /H_2_O (75:25); and flow rate 4 mL/min, to produce compound **2** (29.10 mg). Section V was then separated by preparative HPLC using MeOH/H_2_O (55:45) to yield compound **1** (35.00 mg).

### 3.4. Determined Method of Antibacterial and the Bactericidal Activity

Antibacterial assays were performed using standard bacteria including *B. subtilis* ATCC 6051, *S. aureus* ATCC 6538, *E. coli* ATCC 25922, *P. aeruginosa* ATCC 27853, and methicillin-resistant *Staphylococcus aureus* (MRSA). The bacteria were isolated and cultivated using clinical and Laboratory Standards for antimicrobial susceptibility testing [15]. A single colony of bacteria were cultured from a master plate and put into a bottle of 10.0 mL LB. medium, cultured usually until the bacteria were growing well after shaking at 37 °C overnight or 28 °C. The UV absorbance was measured at 600 nm with 1.0 mL of the culture which the bacterial concentration controlled and determined to each well by ultraviolet spectrophotometer. The corresponding OD values of the different strains were respectively at Abs.600 nm as: 

*B. subtilis* ATCC 6051, A = 0.025; *S. aureus* ATCC 6538, A = 0.072; *E. coli* ATCC 25922, A = 0.055; *P. aeruginosa* ATCC 27853, A = 0.054; and MRSA (clinically isolated strain), A = 0.066.

The main purpose of the study was to determine the in vitro inhibitory and bactericidal activities and the level of tolerance to the three compounds observed by standardized MIC and MBC tests. MICs of each compound were determined by broth two fold serial dilution technique, in accordance with the guidelines of the Clinical and Laboratory Standards Institute [15]. Compounds **1**–**3** were tested at dilution ranges of 1000, 625, 312, 256, 128, 78, 64, 32, 16, 8, 3.12, to 1.56 μg/mL. Standard cation-adjusted penicillin G, gentamycin, and vancomycin were used as positive controls for MIC testing, with 70% methanol for the negative control of each petri dishes. Quantitative 3 μL of the tested sample was added to each well and cultured for 12 h at 37 °C or 28 °C. The plate was complemented with 1.00% tryptone, 0.55 g yeast extract, 15.00 g sodium chloride, adjusted to pH 7.5 and then add 1.50% agar as a semisolid medium for the bacterial detection. MBCs were determined in accordance with the guidelines of the Clinical and Laboratory Standards Institute [18]. The entire volume 100 μL of the MIC well and the wells with 4 dilutions above the MIC were spread across the center of a broth agar plate and allowed to dry for 20 min. Then, a sterile spreading rod was used to evenly disperse the inoculum over the entire surface of the plate, which was then incubated at 37 °C for 24 to 48 h. The MBCs were recorded as the lowest dilution to produce a 99.9% reduction in growth in comparison to the growth of the control. Furthermore, drug resistance can be determined when the tested drug MBC/MIC is at or over 32 times [19,20]. Vancomycin and ticlonin have been demonstrated. Vancomycin lacked bactericidal activity (defined as an MBC/MIC ratio of > or = 32) against two methicillin-resistant *Staphylococcus aureus* (MRSA) isolates from patients with bone and joint infection [21,22]. By contrast, MIC values of many strains were consistent with the tolerance in literature reported (MBC/MIC ratios ≥ 32) [23].

### 3.5. Molecular Docking Study

Compounds **1** and **2** belong to the 5,22*E*-cholestadienol derivatives have the broad-spectrum bactericide activity. This work aimed to evaluate the structure–activity relationships and to establish the mode of interaction by receptor–ligand interactions molecular modeling, thus revealing the action mechanism of target molecules. 

Computational docking study was performed using PharmMapper method to detect and identify drug targets by Discovery Studio 3.0 (East China University of Science & Technology, and Shanghai institute of medicine, Chinese academy of sciences, Shanghai, China) [24]. Based on the possible targets screened, the target protein 2XRL of the doxycycline was found as the broad-spectrum antibiotics receptor [25] and 1Q23 protein of fusidic acid was the Gram-positive cocci receptor [26,27]. Both antibiotics structural was shown in Figure 8. According to the antibiotic drug doxycycline and the corresponding target protein (PDB ID: 2XRL) binding model, and fusidic acid with target protein (PDB ID: 1Q23) binding model, compounds **1** and **2** were further studied by molecular docking. 

Through some related data mining and receptor ligand interaction verification, drug targets of compounds **1** and **2** can be predicted involving the hydrogen bonds, electrostatic forces, van der Waals forces, etc., and the lead compound with 5,22*E*-cholesterol derivatives structure can be confirmed for the compounds **1** or **2**. The active site of the target protein of two compounds had the broad-spectrum bactericide proteins 2XRL and anti-G+ bacteria 1Q23, which there were PDB site displayed in Figure 9. Regions of active site were functionally defined and confirmed independently through automatically parameters correction of the image processing computer. 

The antibacterial activity of 5,22*E*-cholesteric compounds **1** and **2** was studied by reverse molecular docking. The doxycycline of an antibiotic clinically used in sensitive bacterial infection was used as reference. The 2XRL protein receptor of doxycycline is a homodimer and its catalytic site located in the active site BC4 of multiple areas (Figure 9a). A 3D graphic description of the docking structures for compounds **1** and **2** are shown in Figure 10. 

We can observe the scores ranking after molecular docking on the molecular absolute Energy, Conf. Number, Relative Energy, and LibDock in electrostatic 2D structure (Table 4). The docking study showed that compound **2** presented a good score (98.2142), which is close to the score of the reference doxycycline control (99.0843). These results are in concordance with that obtained on in vitro assays (Table 3). Actual antimicrobial activity of compound **2** is stronger than compound **1**. A 2D graphic description of the ligand and receptor protein interactions is shown in Figure 11.

Although these new compounds are not as strong as the effective antimicrobial agents doxycycline, we made the hypothesis that compounds **1** and **2** could be the bactericide overcoming the issues of bacterial resistance through covalent binding interactions between their binding site and the receptor protein. The interaction between each compound and the key amino acids residues of the active site of 2XRL can be clearly seen from 2D diagram of chemical bonding interaction (Figure 11). Molecular docking of compounds showed important van der Waals and hydrogen binding interaction with active pocket of amino acid residues as GlnA:109 and SerA:135 from doxycycline receptor proteins; compound **1** displayed the hydrogen binding interaction with GlnA:109 and van der Waals force; however, there were residues inthe SerA:135 hydrogen binding and π–π interaction of PheA:86 in compound **2**, which would be a good explanation that 5,22*E*-cholestadienol compounds have the broad-spectrum of antibacterial activity and the different levels of inhibition are related to the interaction strength of those compounds.

Secondly, based on the evaluation of the same skeleton antibiotic fusidic acid from PDB ID: 1Q23 (Figure 9b), the target molecular docking was used to demonstrate mechanisms of 5, 22E-cholestadienol compounds **1** and **2** against Gram-positive bacteria *S. aureus* ATCC 6538. The receptor protein 1Q23 of Fusidic acid bond site with amino acid residues for active pocket or catalytic site is located on the active site AC1 of multiple areas (Figure 9b). A 2D graphic description of the docking fusidic acid as a control for compounds **1** and **2** is shown in Figure 12. The docking site and binding of fusidic acid was selected as reference.

Compounds **1** and **2** had scores of 124.669 and 125.637, respectively (Table 5). These scores reveal that the chemical binding force of the extract substances is superior to the score of antibiotics fluidic acid (122.250). These results are in concordance with those obtained on the in vitro assays (Table 3). A 2D graphic description of the ligand interactions are shown in Figure 12.

Table above indicates that compounds **1** and **2** are very strong antimicrobial agents even compared to a positive control. Based on the receptor–ligand interactions of compounds **1**, **2** and the control with the active site of target protein 1Q23, the antibacterial activities and mechanism of action in the 2D diagram of chemical bonding interaction are evident (Figure 13). Molecular docking of compounds showed important Van der Waals and hydrogen binding interaction with active pocket of amino acid residues as SerG:146 and ThruG:172 between fusidic acid and receptor proteins 1Q23; compound **1** displayed the stranger hydrogen binding interaction with HisH:193 Sigma–Pi interaction and Van der Waals force; compound **2** had similar Van der Waals force and Sigma–Pi interaction of TyrJ:33 with the amino acid residues. Therefore, compound **2** is another strong Gram-positive bactericide according to the discovery of molecular docking mechanism, which explains the strong bactericidal activity against *S. aureus* ATCC 6538 of compound **2**, which was observed in the previous antibacterial experiments with MICs of 16 µg/mL and MBCs of 32 µg/mL.

## 4. Conclusions

Currently, there is great interest in the study of insects and related arthropods to develop new natural products for various medicinal purposes. In this work, we found three new compounds from the crude extract of scorpion *Buthus martensii* karsch. Among them, compounds **1** and **2** of the 5,22*E*-cholestadienol derivatives were identified as broad-spectrum bactericides, and their activity showed significant concentration-dependent effects against *Staphylococcus aureus* ATCC 6538 and *Pseudomonas aeruginosa* ATCC 27853. 

Compound **2** was found to fit well in the active site of binding pocket both target proteins (PDB ID:2XRL and PDB ID:1Q23) from the chemical bonds formed intermolecular forces. This molecular docking study demonstrated that compound **2** is an effective lead compound for natural antibiotics, whether it is as a broad-spectrum antibiotic potence. The antibacterial mechanism is the specific binding (various of bonding forces between molecules) using compound **2** as a ligand based on the different receptor proteins 2XRL or 1Q23 active sites from bacterial ribosome unit A, and thus prevent the synthesis of bacterial proteins. This unique mechanism avoids the cross-resistance issues of other antibacterial drugs.

In the present stereochemical studies, we established molecular configuration of new compounds **1** and **3**. The absolute configurations of compound **1** were confirmed by comparing the known chiral analogs of C-24R orientation (24α for the ethyl group) with its *J* value analysis of chemical correlation between H-17ax and H-20eq in the ^1^H-NMR spectrum of compound **1**. Similarly, the absolute configuration of compound **3** proposed the α-bisaboleneol structure as a reference on the Raharivelomanana’s group, and we report the stereochemical determination of compound **3**. Furthermore, The CD spectra data of compounds **1**–**3** also fully support the stereochemical assignments. 

## Figures and Tables

**Figure 1 molecules-24-00072-f001:**
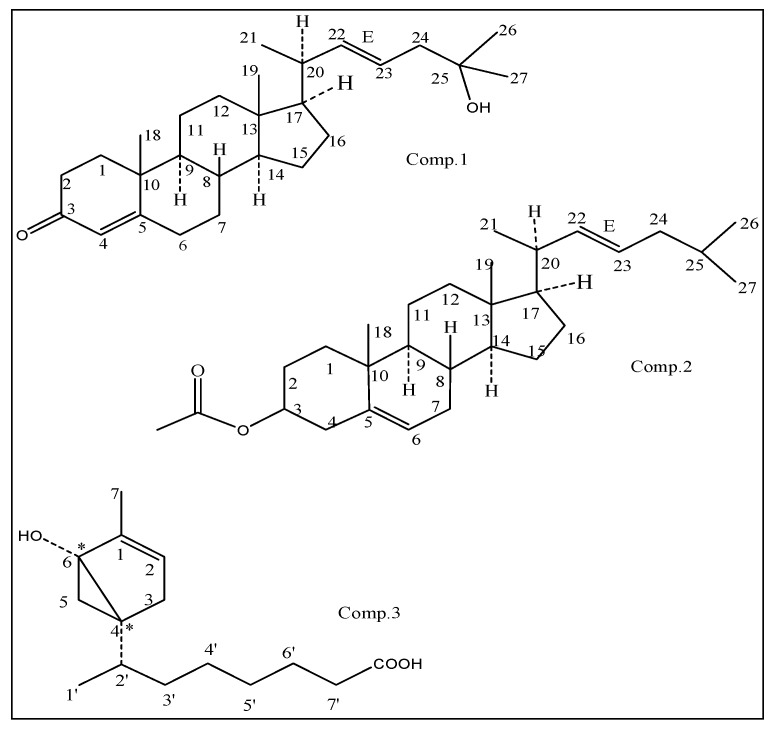
Structures of compounds **1**–**3** from arthropod scorpion, *B. martensii* kirsch.

**Figure 2 molecules-24-00072-f002:**
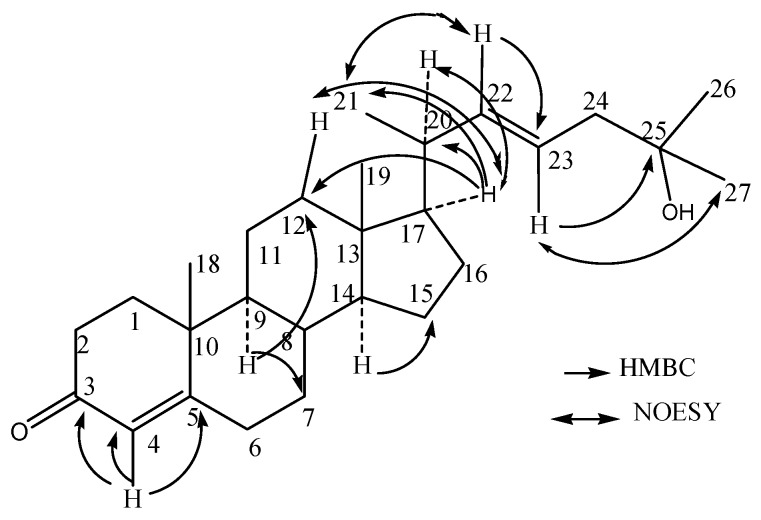
The key HMBC and NOESY correlations of compound **1**.

**Figure 3 molecules-24-00072-f003:**
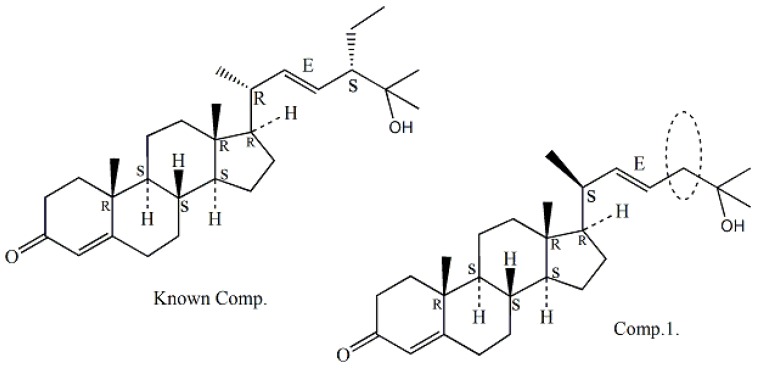
Structure of CAS No. 221012-56-2 and compound **1**.

**Figure 4 molecules-24-00072-f004:**
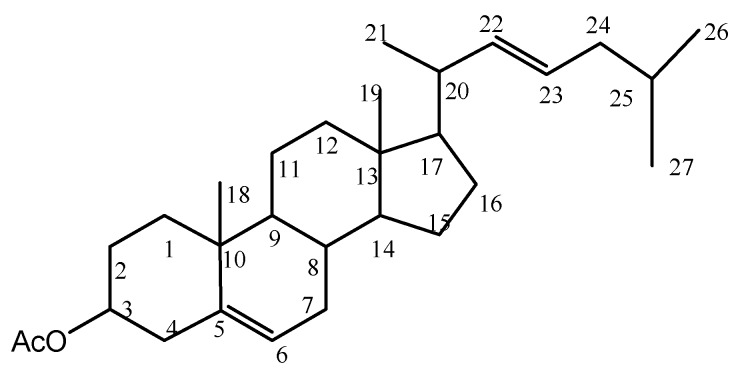
Structure of isolated compound **2**.

**Figure 5 molecules-24-00072-f005:**
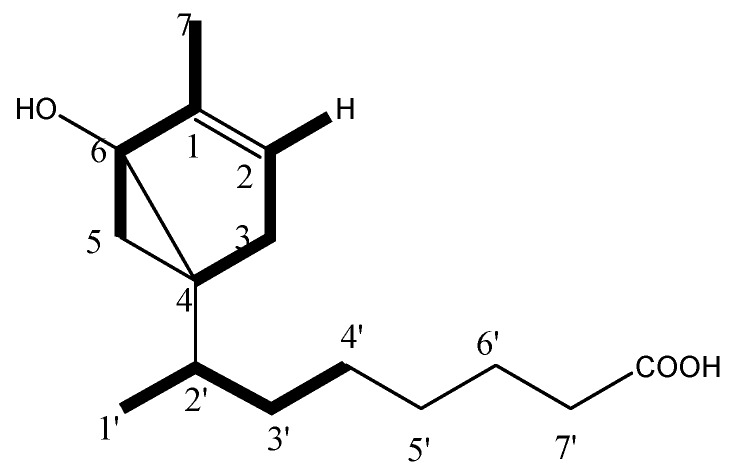
Key ^1^H-^1^H Cosy correlations of compound **3**.

**Figure 6 molecules-24-00072-f006:**
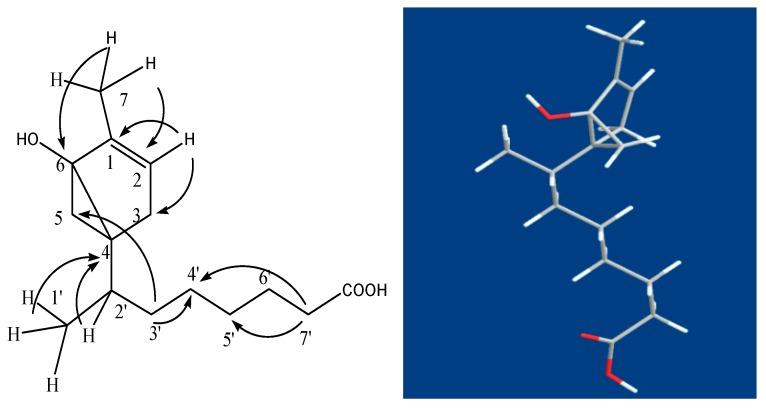
Key HMBC correlations of compound **3**.

**Figure 7 molecules-24-00072-f007:**
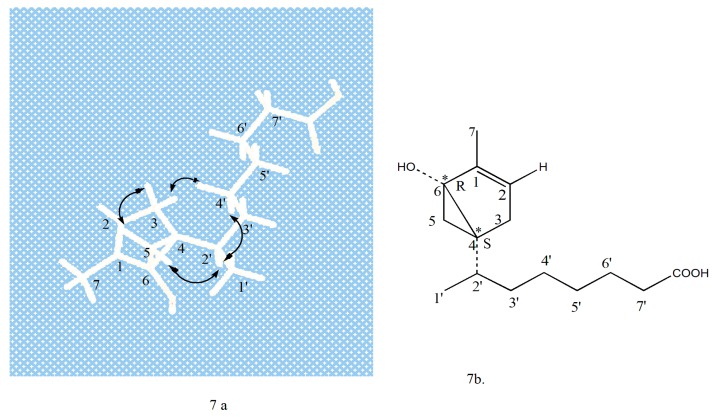
The absolute configuration of compound **3**. (**a**) NOESY correlations of compound **3**; (**b**) The absolute configuration of compound **3** (The asterisk represents the chiral carbon atom).

**Figure 8 molecules-24-00072-f008:**
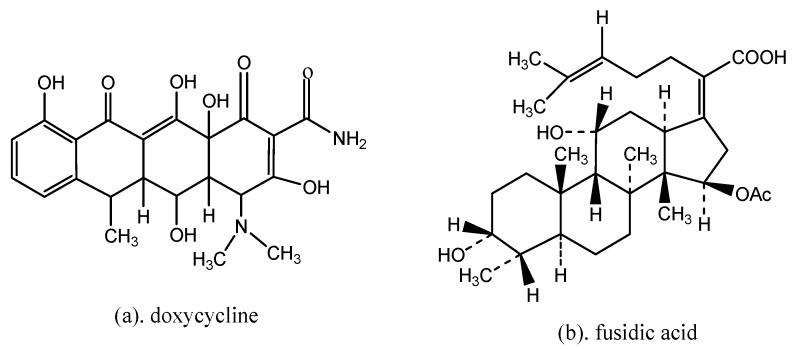
The structural of positive control drugs (**a**,**b**). (**a**) doxycycline (broad-spectrum antibiotic) and (**b**) fusidic acid (antibiotic of Gram-positive cocci infection).

**Figure 9 molecules-24-00072-f009:**
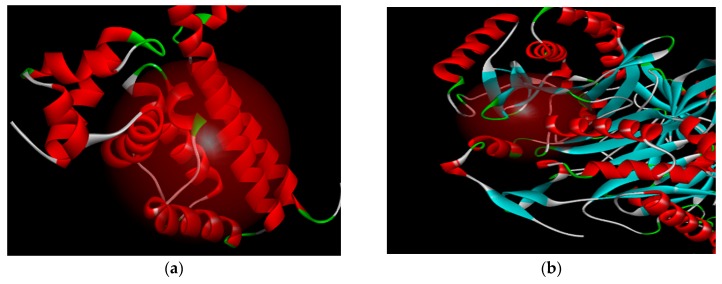
The red globular region in the figure shows the location of the target protein active site. (**a**) Active site of target protein 2XRL (target protein receptor of doxycycline) and (**b**) active site of target protein 1Q23 (target protein receptor of Fusidic acid).

**Figure 10 molecules-24-00072-f010:**
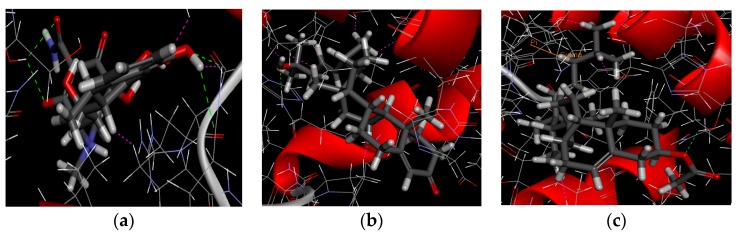
3D docking of ligand–receptor interactions with doxycycline as a control. (**a**) The graphical representation of 3D interaction of doxycycline and receptor protein 2XRL; (**b**) the graphical representation of 3D interaction of compound **1** and receptor protein 2XRL; and (**c**) the graphical representation of 3D interaction of compound **2** and receptor protein 2XRL.

**Figure 11 molecules-24-00072-f011:**
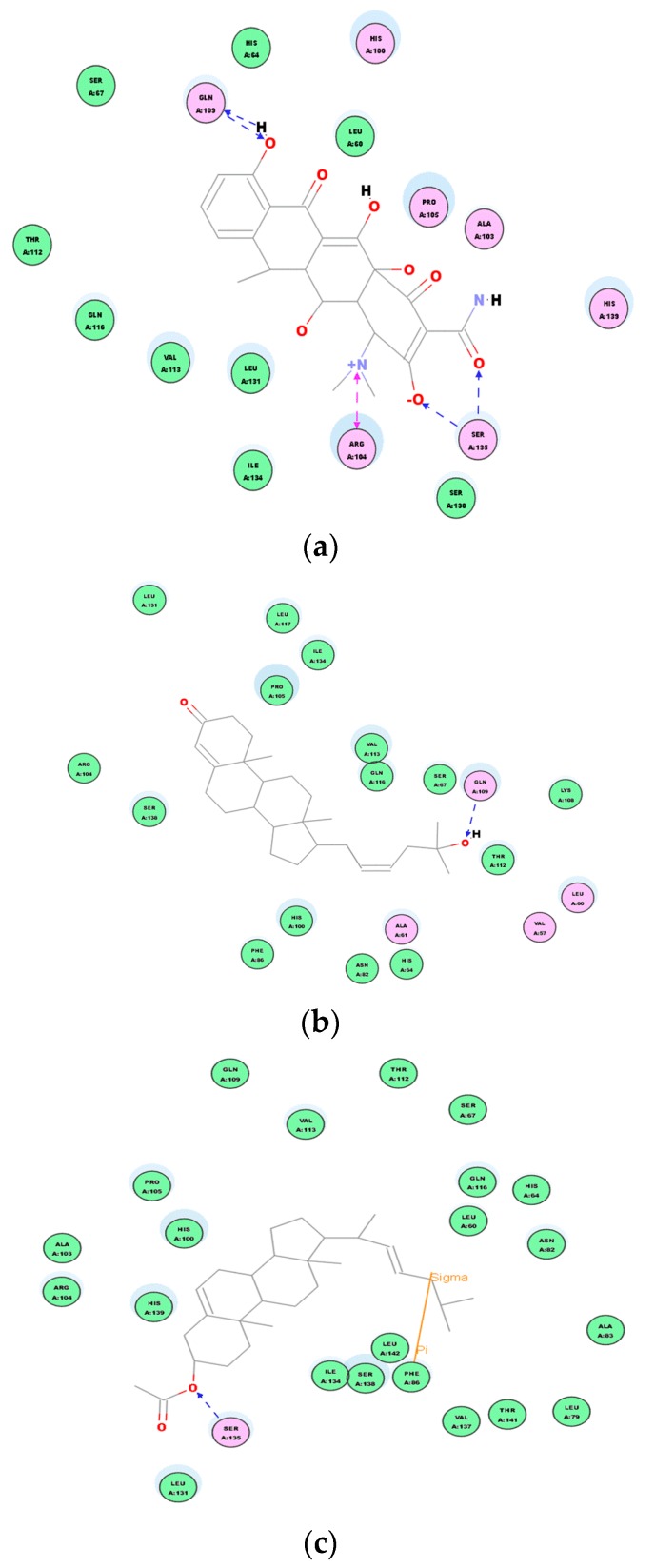
Ligand interaction of compounds **1**, **2** in the active site of target protein 2XRL. (**a**) 2D intermolecular forces and interaction of doxycycline and receptor protein 2XRL. (**b**) 2D intermolecular forces interaction of compound **1** receptor protein 2XRL. (**c**) 2D intermolecular forces interaction of compound **2** receptor protein 2XRL.

**Figure 12 molecules-24-00072-f012:**
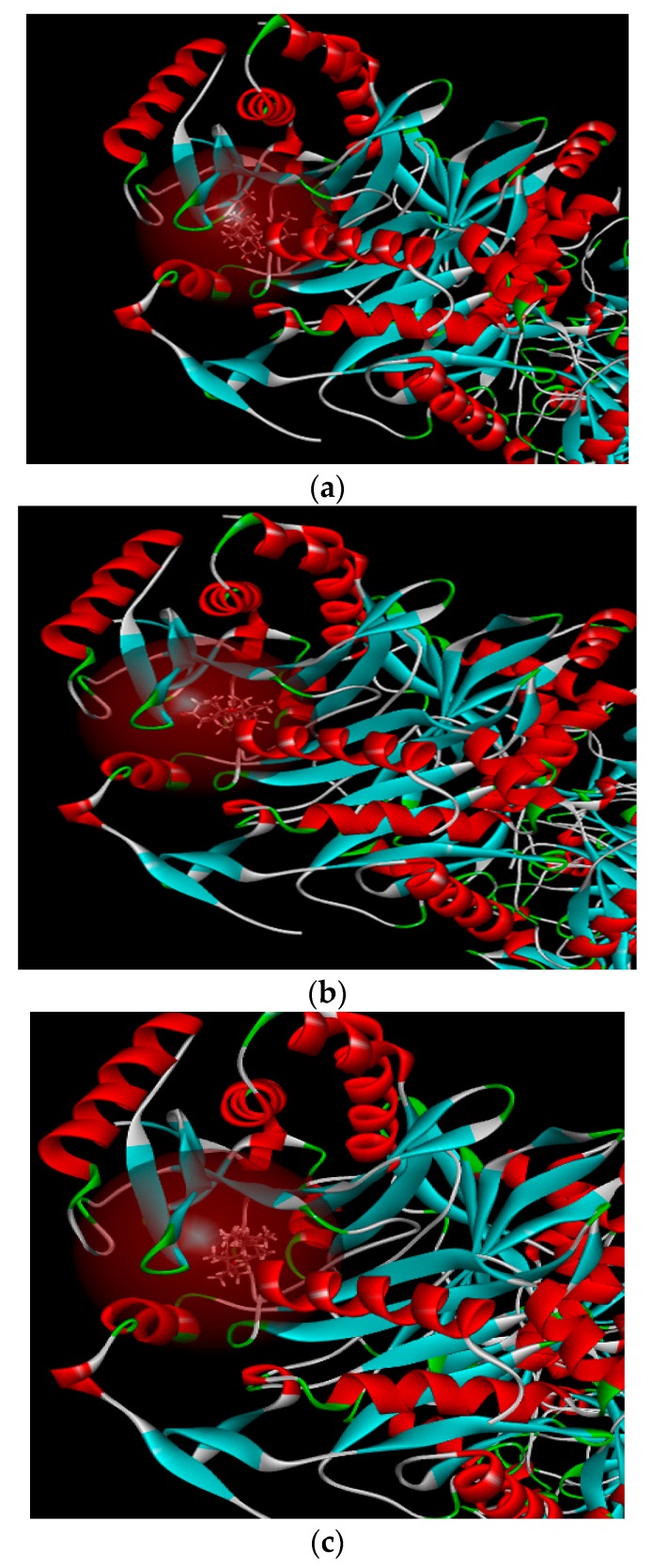
2D docking dimensions of ligand–receptor interactions with fusidic acid as a control. (**a**) 2D interaction models of fusidic acid and receptor protein 1Q23. (**b**) 2D interaction models of compound **1** and receptor protein 1Q23. (**c**) 2D interaction models of compound **2** and receptor protein 1Q23.

**Figure 13 molecules-24-00072-f013:**
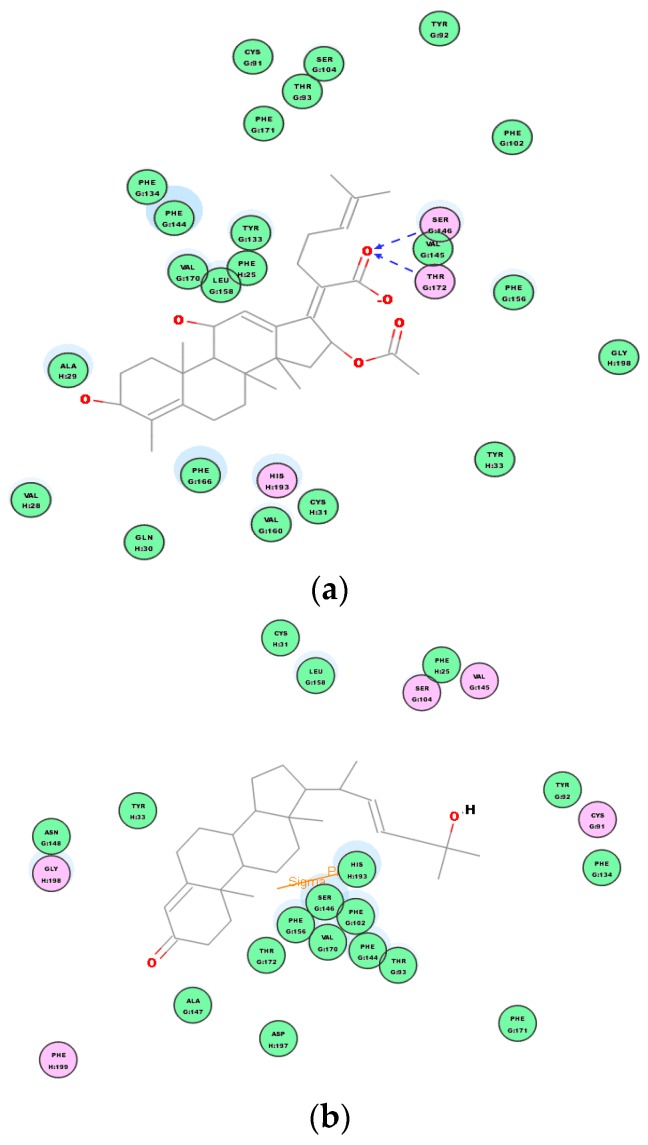
The chemical bonds and intermolecular forces of ligand compounds **1** and **2** in the active site of target protein 1Q23. (**a**) 2D intermolecular forces interaction of fusidic acid and receptor protein 1Q23. (**b**) 2D intermolecular forces interaction of compound **1** and receptor protein 1Q23. (**c**) 2D intermolecular forces interaction of compound **2** and receptor protein 1Q23.

**Table 1 molecules-24-00072-t001:** ^1^H-NMR (400 MHz) and ^13^C-NMR (100 MHz) data of compounds **1** and **2** (in CDCl_3_).

No.	Compound 1		Compound 2	
	δ*_H_* (*J*, Hz)	δ*c* (DEPT)	δ*_H_* (*J*, Hz)	δ*c* (DEPT)
1	1.57 m, 1.20 m	38.3 (CH_2_)	1.02 m, 2.32 m	36.2 (CH_2_)
2	2.24 (2H, m)	35.7 (CH_2_)	1.86 m, 1.55 m	27.2 (CH_2_)
3	-	202.3 (C)	3.50 m	71.8 (CH)
4	5.69 (1H, d, 1.48)	126.1 (CH)	1.82 m	33.1 (CH_2_)
5	-	165.1 (C)		140.7 (C)
6	2.39 m	38.7 (CH_2_)	5.33 (1H, dd, 3.56, 1.52)	121.7 (CH)
7	1.02 m, 1.93 m	31.2 (CH_2_)	2.27 m, 1.25 m	31.4 (CH_2_)
8	1.60 m	29.7 (CH)	1.56 m	31.9 (CH)
9	0.91 m	45.4 (CH)	0.90 m	50.1 (CH)
10	-	42.3 (C)	-	36.2 (C)
11	1.50 m, 1.33 m	21.2 (CH_2_)	1.50 m, 1.06 m	21.1 (CH_2_)
12	1.18 m, 1.96 m	39.5 (CH_2_)	1.30 m, 2.30 m	39.7 (CH_2_)
13	-	50.0 (CH)	-	56.1 (CH)
14	0.90 m	42.8 (CH)	2.33 (1H, m)	42.3 (CH)
15	1.26 m, 2.40 m	23.8 (CH_2_)	1.30 m, 2.30 m	24.7 (CH_2_)
16	1.36 m, 2.02 m	28.1 (CH_2_)	1.86 m, 1.49 m	28.5 (CH_2_)
17	1.58 (1H, dd, 3.78, 1.36)	54.8 (CH)	1.16 m	56.7 (CH)
18	0.84 s	12.0 (CH_3_)	0.84 s	11.9 (CH_3_)
19	0.78 s	17.3 (CH_3_)	0.67 s	18.7 (CH_3_)
20	1.18 (1H, dd, 3.78, 1.48)	36.3 (CH)	1.06 m	37.3 (CH)
21	0.86 s	18.9 (CH_3_)	0.97 s	18.9 (CH_3_)
22	5.68 (1H, dd, 3.24, 1.48)	131.9 (CH)	5.35 (1H, dd, 3.56, 1.64)	131.7 (CH)
23	5.20 (1H, dd, 7.60, 3.28)	135.6 (CH)	5.16 (1H, dd, 6.68, 3.56)	135.8 (CH)
24	1.33 m, 2.40 m	39.5 (CH_2_)	1.12 m, 2.32 m	29.7 (CH_2_)
25	3.67 (1H, m)	70.5 (C)	2.27 (1H, m)	28.5 (CH)
26	0.87 s	22.7 (CH_3_)	1.03 (3H, d, 6.9)	22.5 (CH_3_)
27	0.90 s	22.8 (CH_3_)	1.06 (3H, d, 6.9)	22.8 (CH_3_)
-OCO	-		-	179.0 (C)
-CH_3_	-		1.08 (s)	28.0 (CH_3_)

**Table 2 molecules-24-00072-t002:** ^1^H-NMR (400 MHz) and ^13^C-NMR (100 MHz) data of compound **3** (in CDCl_3_).

No.	δ*_H_* (*J*, Hz)	δ*_c_* (DEPT)	HMBC
1		130.0 (C)	
2	5.34 (1H, br. s)	129.7 (CH)	H-2/C-3, C-1
3	2.01 (2H, br. s)	27.2 (CH_2_)	H-3/C-2, C-2′,
4		29.3 (C)	
5	1.62 (2H, br. s)	24.9 (CH_2_)	
6		77.2 (C)	H-7/C-6
7	0.87 (3H, s)	14.0 (CH_3_)	H-7/C-3, C-2
1′	0.88 (3H, s)	14.1 (CH_3_)	
2′	1.26 (1H, br. s)	29.1 (CH)	H-2′/C-4
3′	2.30 (2H, br. s)	34.1 (CH_2_)	H-3′/C-5, C-4′
4′	1.26 (2H, br. s)	32.0 (CH_2_)	H-4′/C-1′
5′	1.26 (2H, br. s)	29.5 (CH_2_)	
6′	1.26 (2H, br. s)	29.7 (CH_2_)	
7′	1.26 (2H, br. s)	22.7 (CH_2_)	H-7′/C-5′, C-4′
-COOH	7.50 (1H)	179.1 (C)	

**Table 3 molecules-24-00072-t003:** Determination of minimal inhibitory concentration and minimal bactericidal concentration of compounds **1**–**3** from scorpion, *Buthus martensii* karsch (μg/mL).

Strains	Compound 1		Compound 2		Compound 3			Countrol	
	MIC	MBC	MIC	MBC	MIC	MBC	Penicillin G	Gentamycin	Vancomycin
*B. subtilis* (ATCC 6051)	78	>128	78	>128	256	>256	-	-	-
*S. aureus* (ATCC 6538)	78	>128	16	32	>256	>256	-	-	-
*E. coli* (ATCC 25922)	256	>256	256	>256	256	>256	-	6	-
*P. aeruginosa* (ATCC 27853)	64	>78	16	<48	>256	>256	-	6	-
*MRSA* (Clinical isolated)	256	>256	128	256	>256	>256	-	-	0.5

“-” in the table means the measurement is meaningless.

**Table 4 molecules-24-00072-t004:** The docking parameter table of ligands and the receptor protein 2XRL.

Ligand Name	Absolute Energy	Conf Number	Relative Energy	LibDock Score
Doxycycline	76.5451	2	2.29878	99.0843
Compound **1**	50.3676	38	10.4028	94.1788
Compound **2**	54.5212	55	9.67329	98.2142

**Table 5 molecules-24-00072-t005:** The docking parameter table of ligand and the receptor proteins 1Q23.

Ligand Name	Absolute Energy	Conf Number	Relative Energy	LibDock Score
Fusidic acid	92.4881	51	13.1839	122.250
Compound **1**	60.4528	81	15.6049	124.669
Compound **2**	47.9485	22	7.98374	125.637

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
