# Peer review of "Neo-5,22E-Cholestadienol Derivatives from Buthus martensii Karsch and Targeted Bactericidal Action Mechanisms"

_molecules, 2018, doi:10.3390/molecules24010072_

Round 1
Reviewer 1 Report
Dear Authors,
The manuscript ID: molecules-392641 entitled: “Neo-5,22E-cholestadienol derivatives from Buthus martensi karsch and targeted bactericide action mechanisms” by Biyu Lv, Weiping Yin, Jiayu Gao, Huaqing Liu, Kun Liu, Jie Bai and Qiangqiang Yang is an interesting paper.
These findings gives hope for newer and enhanced therapeutic agents derived from ancient arthropod to combat infectious diseases. Such antimicrobials can potentially be translated into clinical practice, however intensive research are needed over the next several years to realize these expectations.
I have some suggestions in order to improve paper, which are the following:
1) major:
- In the whole text are different reference strains???
Please explain this fact:
L. 19-20: S. aureus ATCC25923, P. aeruginosa ATCC27853
L. 206-207 and Table 3: Bacillus subtilis ATCC 6051, Staphylococcus aureus ATCC 6538, Escherichia coli ATCC 25922 and Pseudomonas aeruginosa ATCC 27853
L. 213-214: P. aeruginosa ATCC27853, S. aureus ATCC6538
L. 277-278: B. subtilis ATCC6633, S. aureus ATCC6538, E. coli ATCC27853, P. aeruginosa ATCC15442
L. 286-287: B. subtilis (ATCC 6633), S. aureus (ATCC 6538), E. coli (ATCC 27853), P. aeruginosa (ATCC15442)
L. 294-295: P. aeruginosa ATCC 27853, S. aureus ATCC 6538.
L. 300-302: P. aeruginosa ATCC27853, against S. aureus ATCC 6538, P. aeruginosa ATCC27853, S. aureus ATCC6538
- Material and Methods: Please describe the methodology for assessing antibacterial and bactericidal activity more accurately. It is not clearly presented. What was the range of concentrations tested compounds?
- Results: What the symbol “–“ in the Table 3 means? No information are given regarding the MFC/MIC ratios of new compounds.
2) minor:
Please:
- correct the whole text and tables for spaces, names and punctuation (e.g. spaces between ATCC and reference strain number, concentations – 16 μg/mL, 48 μg/mL, etc.)
- write the full definition and names in the first mention.
- prepare tables and descriptions of tables in accordance with the instructions for authors.
- align and add descriptions to figures: 9, 10, 12.
- unify in the whole text (space, uppercase or lowercase letters) – 5,22E-cholestadienol or 5, 22E-cholestadienol or 5,22E-Cholestadienol – ???
The another details revision has been made directly in the attached manuscript pdf file.
Best regards,

Author Response
Dear Editor:
Thank you very much for your useful comments and suggestions (Open Review 1 and Open Review 2) on the our manuscript (Ref. molecules-392641). Based on these comments and suggestions, we have revised the manuscript carefully. All changes made in the text are marked with blue color in the revised version, and the detailed corrections are listed below point by point. Please let me know if you have any questions.
The revision has been resubmitted to your journal. We look forward to your positive response.
Sincerely,
Prof. Weiping Yin

Reviewer 2 Report
The manuscript describes a study characterizing the antibacterial molecules isolated from scorpion. The structures of bioactive compounds were elucidated by 2D-NMR and HRESI-MS techniques. Molecular docking analysis was performed to identify the potential targets.
The findings reported in this study may benefit the related research area, however, the quality of this manuscript was compromised by several critical issues. Several grammatical and technical errors found throughout the manuscript should be eliminated and paragraphs written in non-scientific style should be revised. The logic, flow and clarity need to be further improved. Extensive editing of English language and style is also required. Several technical issues in the materials and methods should be addressed and clarified. In addition, the authors should further expand the depth and breadth of the discussion. Due to the poor graphic quality, several figures are not acceptable for publication. Due to the problematic issues and poor quality of the manuscript, this manuscript should not be accepted for publication in Molecules.
The specific comments for this manuscript:
1. Line 13 Abstract:
“The powerful weapon against bacteria could be a kind of new antimicrobial molecules” was written in non-scientific style. Revision is recommended.
2. Line 15 “we had been reported the secondary…” grammatical error should be corrected
3. Line 19-20 The abbreviation MBCs and MICs should be spelt out when they were mentioned first time.
4. The paragraph “The antibacterial mechanism is that specific binding with unit A of the 22 bacterial ribosome to interfere and prevent the synthesis of bacterial proteins by molecular docking 23 of compound 2 revealed” is very confusing and was written in non-scientific style. Revision is recommended.
5. Line 33 “effects of Scorpion toxin…” should be lower case ““effects of scorpion toxin…”
6. Line 33-34 grammatical errors found for the paragraph “Insects and arthropod secondary metabolites are an important source of biologically active and therapeutically relevant small molecules…”. Revision is recommended.
7. Line 35-37 grammatical errors found for the paragraph “They have adapted under some of Earth’s harshest living conditions such as mosquito, flies and bedbugs, etc. and responded to environmental challenges by producing biologically active secondary metabolites”. Revision is recommended.
8. Line 38-42 several grammatical errors found in this section. Revision is recommended.
9. Line 42 “This recent study was investigated the antibacterial effects of the scorpion” This paragraph was written in non-scientific style. There are several grammatical errors found in this section.
10. Line 45 There are several grammatical errors found in this section. For example, “Three new compounds were reported and two of them were bactericide ability higher which deduced as the new novel” .
11. Figure 1 The graphic quality for the Fig 1 is unacceptable.
12. Line 52-56 Should be described in Materials and Method, not in Results and Discussion.
13. Figure 3 The graphic quality for the Fig 1 is unacceptable.
14. Line 112-115 grammatical errors
15. Line 208 “Two well-known antibiotics Penicillin G …” was written in non-scientific style. Revision is recommended.
16. Line 211 grammatical error for the statement “have a broad’s antibacterial spectrum….”
Results and Discussion:
The authors focused >95% of the efforts on presenting the results. The authors should further expand the depth and breadth of the discussion.
Materials and Methods
Line 237 grammatical errors found in the paragraph “IR spectra were recorded on a JASCOFT-IR 620 spectrophotometer and UV spectra on a 237 UV-2600 instrument.”
Line 239-240 grammatical and technical errors. “ESI-MS and HR-ESI-MS were obtained” should be revised to “The mass spectra were obtained ….”
Line 242-250 several grammatical errors were found throughout this section.
Line 261-274 This section was written in non-scientific style and several grammatical errors were found.
Line 292-305 “The results of drug sensitivity test showed that 292 Compound 1-2…” should not be part of the Materials and Methods. They should be in the Results and Discussion
Line 307-315 This section was written in non-scientific style and several grammatical errors were found.
Figure 9,10, 11 Poor graphic quality. They are not acceptable.
Several grammatical and technical errors found throughout the manuscript should be eliminated and paragraphs written in non-scientific style should be revised. Extensive editing of English language and style is also required.
Author Response

(The authors gave the same response as above.)

Round 2
Reviewer 1 Report
Dear Authors,
The manuscript ID: molecules-392641-v2 entitled: “Neo-5,22E-cholestadienol derivatives from Buthus martensi karsch and targeted bactericide action mechanisms” by Biyu Lv, Weiping Yin, Jiayu Gao, Huaqing Liu, Kun Liu, Jie Bai and Qiangqiang Yang has not been enough corrected according to review's suggestions.
Please revise the whole text and tables for spaces, names and punctuation (e.g. spaces between ATCC and reference strain number (ATCC6538 or ATCC27853, etc. – ATCC 6538 or ATCC 27853, etc.), concentrations – 16 μg/mL, 48 μg/mL, 32 μg/mL), etc.
bactericide activity – please correct on bactericidal activity
germicide activity – please correct on germicidal activity, but the proper definition is bactericidal activity
No information are given regarding the MBC/MIC ratios and bactericidal or bacteriostatic effects of 5,22E-cholestadienol derivatives; MBC/MIC ≤ 4 – bactericidal effect, MBC/MIC > 4 – bacteriostatic effect.
L. 197, 281: staphylococcus aureus – Staphylococcus aureus
L. 202: MICs – MICs (minimal inhibitory concentrations) – Please write the full definition in the first mention
L. 208: minimal bactericide concentration – minimal bactericidal concentration
L. 301-302: “Furthermore, drug 301 resistance can be determined when the tested drug MBC /MIC is at or over 32 times” – please add the position of the literature
Best regards,
Author Response
Responses to review 1 (Round 2)
Dear reviewer :
Thank you very much for your Round 2 of review’s comments (Open Review 1 and Open Review 2) on the our manuscript (Ref. molecules-392641). Based on these comments and suggestions, we have revised the manuscript carefully. All changes made in the text are marked with blue color in the revised version, and the detailed corrections are listed below point by point. Please let me know if you have any questions.
The revision has been resubmitted to your journal. We look forward to your positive response.
Sincerely,
Prof. Weiping Yin
Please revise the whole text and tables for spaces, names and punctuation (e.g. spaces between ATCC and reference strain number (ATCC6538 or ATCC27853, etc. – ATCC 6538 or ATCC 27853, etc.), concentrations – 16 μg/mL, 48 μg/mL, 32 μg/mL), etc.
√ According to the review’s suggestion, they have been modified in our revised version.
bactericide activity – please correct on bactericidal activity
√ According to the review’s suggestion, it has been modified in our revised version
germicide activity – please correct on germicidal activity, but the proper definition is bactericidal activity
√ According to the review’s suggestion, this vocabulary has been revised uniformly and changed into“bactericidal activity”throughout the text, based on the original definition.
No information are given regarding the MBC/MIC ratios and bactericidal or bacteriostatic effects of 5,22E-cholestadienol derivatives; MBC/MIC ≤ 4 – bactericidal effect, MBC/MIC > 4 – bacteriostatic effect
√ According to the review’s suggestion, the content of this aspect has been supplemented in the second paragraph of the 2.2 Assay of antibacterial and bactericidal activities (line 217), and relevant references have been added. As follows:
5,22E-cholestadienol derivatives (compound 2 and 1) are small molecules of natural products with the significant inhibitory effects against pathogenic bacteria S. aureus ATCC 6538 and P. aeruginosa ATCC 27853. And MBC/MIC ratios of two compounds were quantified using a luciferase-based assay [16]. According to the ratio of MBC/MIC, it is possible to identify the antibacterial profile of compounds (bacteriostatic and/or bactericidal). The result shows that compound 2 inhibited bacterial growth of both S. aureus and P. aeruginosa in a bactericidal rather than a bacteriostatic manner (MBC/MIC ratio≤2). Similarly, with compound 1, a ratio of MBC/MIC ≤2 indicates bactericidal activity inhibited bacterial growth of P. aeruginosa, whereas a ratio of MBC/MIC ≥4 defined a bacteriostatic effect. [17] Interestingly, this suggests that these two compounds can be classified as bactericidal agents against broad spectrum bactericide activities of 5,22E-cholestadienol derivatives from Buthus martensii karsch, which is a more interesting profile.
16. Nilda, V. A. Núñez; Humberto H. L. V.; Liliana del C.I. T.; Cristina R. P. Silver Nanoparticles Toxicity and Bactericidal Effect Against Methicillin-Resistant Staphylococcus aureus:Nanoscale Does Matter. Nanobiotechnol, 2009,1-9. DOI 10.1007/s12030-009-9029-1
17. Konaté, K. ; Mavoungou,J. F.; Lepengué, A. N.; Samseny, R. R. R. A.; Hilou, A.; Souza, A.; Dicko M. H.; M'Batchi, B. Antibacterial activity against β- lactamase producing Methicillin and Ampicillin-resistants Staphylococcus aureus: Fractional Inhibitory Concentration Index (FICI) determination. Ann. Clin. Microbiol. Antimicrob., 2012, 11, 18.
L. 197, 281: staphylococcus aureus – Staphylococcus aureus
√ According to the review’s suggestion, it has been modified in our revised version
L. 202: MICs – MICs (minimal inhibitory concentrations) – Please write the full definition in the first mention
√ According to the review’s suggestion, it has been modified in our revised version
L. 208: minimal bactericide concentration – minimal bactericidal concentration
√According to the review’s suggestion, it has been modified in our revised version
L. 301-302: “Furthermore, drug resistance can be determined when the tested drug MBC /MIC is at or over 32 times” – please add the position of the literature
√ According to the review’s suggestion, this sentence has been supplemented and added relevant references. It has been changed into “Furthermore, drug resistance can be determined when the tested drug MBC/MIC is at or over 32 times [19,20]. Vancomycin and ticlonin have been demonstrated. Vancomycin lacked bactericidal activity (defined as an MBC/MIC ratio of > or = 32) against 2 methicillin-resistant Staphylococcus aureus (MRSA) isolates from patients with bone and joint infection [21,22]. By contrast, MIC values of many strains were consistent with the tolerance in literature reported(MBC/MIC ratios ≥32)[23]..
Reference:
19. May, J.; Shannon, K.; King, A.; French. G. Glycopeptide tolerance in Staphylococcus aureus. J. Antimicrob. Chemother. 1998, 42, 189–197.
20. Maria M.; Traczewski, B. D.; Katz, J. N. S.; Steven , D. B. Inhibitory and bactericidal activities of Daptomycin, Vancomycin, and Teicoplanin against Methicillin-Resistant Staphylococcus aureus isolates collected from 1985 to 2007. Antimicro Agents Chemother. 2009, 53, 1735-1738.
21. Rouse, M.S.; Steckelberg, J.M.; Patel, R. In vitro activity of ceftobiprole, daptomycin, linezolid, and vancomycin against methicillin-resistant staphylococci associated with endocarditis and bone and joint infection. Diagn Microbiol Infect Dis. 2007 , 58, 363-365.
22. Nicholas, S. B.; Nimish P.; Theresa, I. S.; Wissam, I. E.; Atrouni, R. T. H.; Molly E. S. Relationship between vancomycin tolerance and clinical outcomes in Staphylococcus aureus bacteraemia . J. Antimicrob Chemother. 2017, 72, 535-542.
23 . Biedenbach, D.J.; Bell, J.M.; Sader, H.S.; Fritsche T.R.; Jones ,R.N.; Turnidge , J.D. Antimicrobial susceptibility of Gram-positive bacterial isolates from the Asia-Pacific region and an in vitro evaluation of the bactericidal activity of daptomycin, vancomycin, and teicoplanin: a SENTRY Program Report (2003-2004). Int J Antimicrob Agents, 2007,30, 143-149.
The revised version has been resubmitted to your journal. We look forward the response.
Sincerely yours ,
Correspondence author : Weiping Yin
Reviewer 2 Report
The authors have made significant efforts to improve the quality of the manuscript. However, the quality of this manuscript was still compromised by several grammatical and technical errors found throughout the manuscript. In addition, the paragraphs written in non-scientific style should be revised. The logic, flow and clarity need to be further improved. Extensive editing of English language and style is also required. The poor graphic quality of several figures in this manuscript is not acceptable for publication purpose. Due to the problematic issues and poor quality of the manuscript, this manuscript should not be accepted for publication in Molecules until these concerns are addressed.
Line 19-22 “It was found that a strong bactericide activity of compound 2 against…” The grammatical errors should be corrected
Line 24-26 “The antibacterial mechanism is the specific binding between unit A of the bacterial ribosome and interfere, and thus prevent the synthesis of bacterial proteins by molecular docking of compound 2”. This should be revised to improve the flow and clarity.
Line 39-40 Please rephrase “the assessment of active molecular diversity and the identities of majority of these small molecules” to improve the clarity.
Figure 3 Poor graphic quality
Line 201 remove “to be against”
Line 203 Please revise “It then selected those sample with MICs value…”
Line 260 please revise “Four widely used the opportunistic pathogenic including…”
Line 261 It should be “Four” rather than “4” at the beginning of the sentence.
Line 279 Please revise “Antibacterial activities of new compounds 1-3 for strains of bacteria tested and the opportunistic….” to “Antibacterial activities of new compounds 1-3 against the opportunistic…were evaluated….”
Line 312 “Compound 1 and 2 were further study Compound 1 and 2 were further study..” The grammatical errors should be corrected.
Figure 8 (a) and (b) Graphic quality is poor.
Line 322 “ofcompound”?
Line 323 “receptor-ligand interaction verificated involves hydrogen bonds….”. The grammatical error should be corrected.
Figure 9. The graphic quality is unacceptable for publication purpose.
The style of the Figure caption should be consistent throughout the manuscript. For example, 8 (a) vs. 9 a, they are not consistent.
Figure 10 The graphic quality is unacceptable for publication purpose.
Line 327 “automatically correctting the parameters by the image processing computer” It was written in non-scientific style. Revision is recommended.
Line 349 “( Table 4)” should be corrected.
Line 357-359 Pleas elaborate and clarify “same covalent binding region of natural compounds as 5,22E-cholestadienol bactericide may overcome the issues of bacterial resistance through the receptor sites”.
Line 363 “stranger hydrogen binding”?
Line 367 “the differen”. The typo should be corrected.
Figure 11 The graphic quality is unacceptable for publication purpose
Line 375 “1Q23( Figure 8b)”
Figure 13 The graphic quality is unacceptable for publication purpose
The quality of this manuscript was compromised by several grammatical and technical errors found throughout the manuscript. In addition, the paragraphs written in non-scientific style should be revised.
Author Response
Responses to review 2 (Round 2)
Dear reviewer:
Thank you very much for your useful comments and suggestions (Open Review 1 and Open Review 2) on the our manuscript (Ref. molecules-392641). Based on these comments and suggestions, we have revised the manuscript carefully. All changes made in the text are marked with blue color in the revised version, and the detailed corrections are listed below point by point. Please let me know if you have any questions.
The revision has been resubmitted to your journal. We look forward to your positive response.
Sincerely,
Prof. Weiping Yin
Line 19-22 “It was found that a strong bactericide activity of compound 2 against…” The grammatical errors should be corrected
√ According to the review’s suggestion, this sentence has been rewritten and changed into“To explore the antibacterial properties of these new compounds , the result shows that compound 2 inhibited bacterial growth of both S. aureus and P. aeruginosa in a bactericidal rather than a bacteriostatic manner (MBC/MIC ratio≤2). Similarly, with compound 1, a ratio of MBC/MIC ≤2 indicates bactericidal activity inhibited bacterial growth of P. aeruginosa. Remarkably, this suggests that two compounds can be classified as bactericidal agents against broad spectrum bactericide activities for 5,22E-cholestadienol derivatives from Buthus martensii karsch . The structures of compound 1-3 were established……..”
Line 24-26 “The antibacterial mechanism is the specific binding between unit A of the bacterial ribosome and interfere, and thus prevent the synthesis of bacterial proteins by molecular docking of compound 2”. This should be revised to improve the flow and clarity.
√ According to the review’s suggestion, this sentence has been rewritten and changed into“The antibacterial mechanism is the specific binding(various of bonding forces between molecules)using compound 2 or 1 as ligand based on the different receptor proteins 2XRL or 1Q23 active sites from bacterial ribosome unit A, and thus prevent the synthesis of bacterial proteins”
Line 39-40 Please rephrase “the assessment of active molecular diversity and the identities of majority of these small molecules” to improve the clarity.
√ According to the review’s suggestion, this sentence has been rewritten and changed into“The assessment of the diversity of active molecules and the identification of novel structures small molecular remain a continuing challenge“
Figure 3 Poor graphic quality
√ According to the review’s suggestion, Figure 3 have been redrawn in the
revised version.
Line 201 remove “to be against”
√ It has been removed in the revised version.
Line 203 Please revise “It then selected those sample with MICs value…”
√ According to the review’s suggestion, this sentence has been rewritten and changed into“In order to obtain the effective fungicide leading compound and further determine the MBC value, we selected the MIC value of less than 64 μg/mL for sustainable development and testing sample” in the revised version.
Line 260 please revise “Four widely used the opportunistic pathogenic including…”
√ According to the review’s suggestion, this sentence has been rewritten and changed into“In the antibacterial experiment, four selected pathogenic strains and one resistant strain were commonly used in clinic, including two gram- positive bacteria (B. subtilis and S. aureus ), two gram-negative bacteria(E. coli and P. aeruginosa ) and the medicine-resistant strain (Methicillin-resistant staphylococcus aureus MRSA) .”in line 276-277 of the revised version.
Line 261 It should be “Four” rather than “4” at the beginning of the sentence.
√ It has been removed in line 277 in the revised version.
Line 279 Please revise “Antibacterial activities of new compounds 1-3 for strains of bacteria tested and the opportunistic….” to “Antibacterial activities of new compounds 1-3 against the opportunistic …were evaluated….”
√ According to the review’s suggestion, this paragraph has been rewritten and changed into“ Antibacterial assays were performed using standard bacteria including B. subtilis ATCC 6051,S. aureus ATCC 6538, E. coli ATCC 25922, P. aeruginosa ATCC 27853 and Methicillin-resistant staphylococcus aureus (MRSA). The bacteria were isolated and cultivated using clinical and Laboratory Standards for antimicrobial susceptibility testing [15]. A single colony of bacteria were cultured from a master plate and put into a bottle of 10.0 mL LB. medium,……. ”
“The main purpose of the study was to determine the in vitro inhibitory and bactericidal activities and the level of tolerance to the three compounds observed by standardized MIC and MBC tests. MICs of each compound were determined by broth two fold serial dilution technique, in accordance with the guidelines of the Clinical and Laboratory Standards Institute [15], Compound 1,2 and 3 were tested at dilution ranges of 1000, 625, 312, 256, 128, 78, 64, 32, 16, 8, 3.12 to 1.56 μg /mL. Standard cation-adjusted Penicillin G, Gentamycin and Vancomycin were used as positive control for MIC testing, 70% methanol for the negative control of each petri dishes. Quantitative 3 μL of the tested sample was added to each well and cultured for 12 hours at 37 °C or 28 °C. The plate was complemented with 1.00 % tryptone, 0.55 g yeast extract, 15.00 g sodium chloride, adjusted to pH 7.5 and then add 1.50 % agar as a semi solid medium for the bacterial detection. MBCs were determined in accordance with the guidelines of the Clinical and Laboratory Standards Institute (18). The entire volume 100 μL of the MIC well and the wells with 4 dilutions above the MIC were spread across the center of a broth agar plate and allowed to dry for 20 min. Then, a sterile spreading rod was used to evenly disperse the inoculum over the entire surface of the plate, which was then incubated at 37°C for 24 to 48 h. The MBCs were recorded as the lowest dilution to produce a 99.9% reduction in growth in comparison to the growth of the control. ”
Line 312 “Compound 1 and 2 were further study.” The grammatical errors should be corrected.
√ According to the review’s suggestion, this sentence has been rewritten and changed into“According to the antibiotic drug doxycycline and the corresponding target protein ( PDB ID: 2XRL) binding model, and fusidic acid with target protein ( PDB ID: 1Q23) binding model, Compound 1 and 2 were further studied by molecular docking”in the revised version.
Figure 8 (a) and (b) Graphic quality is poor.
√ According to the review’s suggestion, Figure 8 have been redrawn in the
revised version.
Line 322 “ofcompound”?
√ It has been corrected in the revised version.
Line 323 “receptor-ligand interaction verificated involves hydrogen bonds….”. The grammatical error should be corrected.
√ According to the review’s suggestion, this sentence has been rewritten and changed into“receptor-ligand interactions involving hydrogen bonds”in line 331 in the revised version.
Figure 9. The graphic quality is unacceptable for publication purpose.
√ According to the review’s suggestion, Figure 9 have been corrected and improved in the revised version.
The style of the Figure caption should be consistent throughout the manuscript. For example, 8 (a) vs. 9 a, they are not consistent.
√ According to the review’s suggestion, this style of the Figure 8 caption has been corrected and changed into:
(a). doxycycline (Broad-spectrum antibiotics)
(b). fusidic acid (Antibiotic of gram-positive cocci infection)
Figure 10 The graphic quality is unacceptable for publication purpose.
√ According to the review’s suggestion, Figure 10 have been corrected and improved in the revised version.
Line 327 “automatically correctting the parameters by the image processing computer” It was written in non-scientific style. Revision is recommended.
√ According to the review’s suggestion, this sentence has been rewritten and changed into“Regions of active site were functionally defined and confirmed independently through automatically parameters correction of the image processing computer.”
Line 349 “( Table 4)” should be corrected.
√ It has been corrected for Table 4 and Table 5 in the revised version, according to the review’s suggestion,
Line 357-359 Pleas elaborate and clarify “same covalent binding region of as 5,22E-cholestadienol bactericide may overcome the issues of bacterial resistance through the receptor sites”.
√ According to the review’s suggestion, this sentence has been rewritten and changed into“compounds 1 and 2 could be the bactericide overcoming the issues of bacterial resistance through covalent binding interaction between their binding site and the receptor protein.
Line 363 “stranger hydrogen binding”?
√ According to the review’s suggestion, we are sorry that this sentence has been rewritten and changed into“the hydrogen binding” in the revised version.
Line 367 “the differen”. The typo should be corrected.
√ We are sorry that this typo has been corrected in line 386 in the revised version.
Figure 11 The graphic quality is unacceptable for publication purpose
√ According to the review’s suggestion, Figure 11 have been corrected and improved in the revised version.
Line 375 “1Q23( Figure 8b)”
√ We are sorry that this typo has been corrected“1Q23 (Figure 9b)” in line 395 in the revised version.
Figure 13 The graphic quality is unacceptable for publication purpose
√ According to the review’s suggestion, Figure 12 and 13 have been revised version. in the revised version.
The quality of this manuscript was compromised by several grammatical and technical errors found throughout the manuscript. In addition, the paragraphs written in non-scientific style should be revised.
Ö The manuscript has been checked again to avoid any grammatical and technical errors and several paragraphs have been revised properly.
The revised version has been resubmitted to your journal. We look forward the response.
Sincerely yours,
Correspondence author : Weiping Yin